# Dermal injury drives a skin to gut axis that disrupts the intestinal microbiome and intestinal immune homeostasis in mice

Tatsuya Dokoshi [1], Yang Chen[1,2], Kellen J. Cavagnero [1], Gibraan Rahman[2], Daniel Hakim[2], Samantha Brinton[1], Hana Schwarz [1], Elizabeth A. Brown[1], Alan O'Neill[1], Yoshiyuki Nakamura[1], Fengwu Li[1], Nita H. Salzman [3], Rob Knight[2,4,5,6] & Richard L. Gallo [1] ✉

The composition of the microbial community in the intestine may influence the functions of distant organs such as the brain, lung, and skin. These microbes can promote disease or have beneficial functions, leading to the hypothesis that microbes in the gut explain the co-occurrence of intestinal and skin diseases. Here, we show that the reverse can occur, and that skin directly alters the gut microbiome. Disruption of the dermis by skin wounding or the digestion of dermal hyaluronan results in increased expression in the colon of the host defense genes *Reg3* and *Muc2*, and skin wounding changes the composition and behavior of intestinal bacteria. Enhanced expression *Reg3* and *Muc2* is induced in vitro by exposure to hyaluronan released by these skin interventions. The change in the colon microbiome after skin wounding is functionally important as these bacteria penetrate the intestinal epithelium and enhance colitis from dextran sodium sulfate (DSS) as seen by the ability to rescue skin associated DSS colitis with oral antibiotics, in germ-free mice, and fecal microbiome transplantation to unwounded mice from mice with skin wounds. These observations provide direct evidence of a skin-gut axis by demonstrating that damage to the skin disrupts homeostasis in intestinal host defense and alters the gut microbiome.

Different epithelial tissue environments such as the skin, gut, and lung each require that the barrier organ must deploy distinct mechanisms to control microbial growth and limit invasion of the epithelial surface[1]. Disease or injury of these epithelial surfaces results in a disrupted physical and immune barrier that will alter immune homeostasis with resident microbes and may enable some organisms to promote disease[2,3]. This may be limited to the local site, but also can occur at the same time as changes to distant organs. For example, several diseases of the skin and gut frequently occur together, such as atopic dermatitis and food allergy[4–6], or psoriasis and inflammatory bowel disease (IBD)[7–9]. These clinical observations suggest the existence of a functional axis between the gut and the skin. It has been commonly hypothesized that microbes in the gastrointestinal system influence the function of the skin epithelial barrier. Similar inter-organ communication has also been found between the gut, lung, and brain[10–13]. However, although compelling clinical and experimental observations have shown that communication exists between distant organs, relatively little is understood about how these organs communicate.

[1]Department of Dermatology, University of California, San Diego, La Jolla, CA, USA. [2]Department of Pediatrics, University of California, San Diego, La Jolla, CA, USA. [3]Department of Pediatrics, Division of Gastroenterology and Center for Microbiome Research, Medical College of Wisconsin, Milwaukee, WI, USA. [4]Department of Computer Science & Engineering, University of California, San Diego, La Jolla, CA, USA. [5]Department of Bioengineering, University of California, San Diego, La Jolla, CA, USA. [6]Center for Microbiome Innovation, University of California, San Diego, La Jolla, CA, USA. ✉e-mail: rgallo@ucsd.edu

Previous experimental models utilizing mice with skin injury have reported the unexpected finding that the skin can influence function of the gut, a communication pathway opposite from the most common assumption that the gut influence the skin[14]. Wounding of mouse skin was shown to result in the acquisition of increased sensitivity to colitis following oral challenge with dextran sodium sulfate (DSS), or in an experimental model of spontaneous colitis in $IL10^{-/-}$ mice[15]. This model reflected the increase in inflammatory bowel disease observed in human patients with chronic skin inflammatory disorders such as psoriasis. A similar, but more severe phenotype was also observed in a genetic mouse model that promoted the digestion of hyaluronan (HA) in the skin by targeted transgenic overexpression of human hyaluronidase-1 (HYAL1) in the epidermis[15]. Intestinal tissue exposed in this manner to increased circulating fragments of hyaluronan were asymptomatic under control conditions but demonstrated increased inflammation, disruption of the epithelial barrier and development of increased adipose tissue in the colon reflective of "creeping fat" commonly seen in patients with IBD[16,17]. Although these observations provided a potential mechanism for communication from skin to gut, it remained unknown why the release of HA from the skin after injury or inflammation would impart an increased risk of colitis.

In this study, we perform a detailed analysis of the intestine and the fecal microbiome after skin wounding or the expression of hyaluronidase in the epidermis to mimic this aspect of tissue injury. Our data show that the skin can promote a change in the gut microbiome that subsequently alters inflammation in the intestine after a challenge by DSS. These changes in the gut experimentally demonstrate that the skin influences gene expression in the colon, thus providing evidence of a skin to gut immune axis.

## Results

### Hyaluronidase activity in the skin induces expression of *Muc2* and *Reg3* in the colon

To better understand how the skin may influence gene expression of cells in the intestine, single-cell RNA sequencing (scSeq) was performed on the whole colon of mice with a skin-specific intervention; transgenic expression of hyaluronidase-1 (HYAL1) under conditional control of the keratin 14 (K14) promoter (K14/HYAL1). This model recapitulates aspects of skin wounding without inducing immune cell migration or systemic cytokine responses and has been shown to greatly increase susceptibility to colitis following oral administration of dextran sodium sulfate (DSS)[15]. The mechanism responsible for this increased susceptibility to disease of the colon was unknown and is used as a model of the association between human skin inflammation and IBD.

Analysis of scSeq data resolved 17 distinct cell clusters, and several of these cell clusters including epithelial cells and lymphocytes clusters were increased in K14/HYAL1 mice (Fig. 1a and Supplementary Fig. 1a). Based on the differentially expressed genes in each cluster, gene ontology (GO) analysis detected a host defense response in the colon from clusters 0-2,4, 5 in K14/HYAL1 mice (Fig. 1b andS upplementary Fig. 1b, c). Of note, the important intestinal antimicrobial and host defense genes *Reg3b, Reg3g, Muc2*, and anterior gradient 2 (*Agr2*) were increased in intestinal epithelial cells of K14/ HYAL1 mice (Fig. 1c and Supplementary Fig. 1d, e). To validate and further extend this observation, spatial RNA sequencing of the intestine was performed on the entire colon after rolling the tissue from the cecum to the rectum to enable a full-length analysis of diverse regions of the colon (Fig. 1d, e). In this analysis, K14/HYAL1 and control mice were also challenged by ingestion of DSS to evaluate the transcriptional response in the colon following intestinal injury. These data resolved into 18 clusters (Fig. 1f and Supplementary Fig. 2), and mapping of these clusters distinguished transcripts localized in the proximal, transverse, and distal colon (Fig. 1g). Several clusters were differentially expressed between experimental groups of K14/HYAL mice compared to control, with greatest enrichment of clusters 1 (Epithelium), 10 (Epithelium) and 16(Crypt) (Fig. 1h and Supplementary Fig. 3a). Analysis of spatial sequencing data confirmed observations from scRNASeq with *Muc2* highest in Cluster 16 (Supplementary Fig. 3b). Interestingly, even with the substantial inflammatory response induced by DSS in the intestine, K14/HYAL1 mice exhibited the highest expression level of *Muc2* in the absence of DSS treatment. (Fig. 1i, j). Spatial sequencing also showed *Reg3b* and *Reg3g* were increased in the colon of K14/HYAL1 mice (Fig. 1k, l) and present at the highest levels in K14/HYAL1 with or without DSS compared to control mice treated with DSS alone (Supplementary Fig. 3c).

To further assess changes in intestinal host defense that could occur with the increase in expression of *Muc2* and *Reg 3g* mRNA, we next evaluated mucin production in goblet cells[18] and protein expression of *Reg3*. In these analysesanalyzed, both K14/HYAL1 mice and mice with aseptic, full-thickness skin wounds were evaluated to detect the response of the colon to skin. Both K14/HYAL1 mice and mice with skin wounding showed an increase in mucin staining in crypts within the transverse colon compared to controls (Fig. 2a and Supplementary Fig. 4a, b). Reg3 protein expression in the transverse colon was increased as seen by Western blot (Fig. 2b, c), and immunostaining further showed that either K14/HYAL1 mice or wounding of the skin increased expression of *Muc2* and *Reg3g* (Fig. 2d, e). Of note, no change in local immune cell infiltration was observed in the colon after skin wounding (Supplementary Fig. 5a, b). In addition, skin injury of germ-free showed increased mucin and *Reg3g* (Fig. 2f and Supplementary Fig. 4c). This result suggested that the capacity of the skin to induce a change in these intestinal host defense genes was not dependent on the presence of bacteria in the intestine.

### Hyaluronan fragments induce expression of *Muc2* and *Reg3* in colon epithelial cells

Evidence of dermal hyaluronidase activity was observed in lesional psoriasis skin by a local loss of dermal HA (Supplementary Fig. 6), suggesting release of this DAMP occurs in human skin inflammatory processes and supporting the relevance of HA in disorders such as psoriasis that are associated with intestinal disease. Since K14/HYAL mice have a systemic increase in exposure to HA fragments, we next asked if these fragments could directly promote the increased expression of *Muc2* and *Reg3* in culture. Hyaluronan fragments were tested directly by addition to ex-vivo cultures of mouse colon tissue and on the human colon epithelial cell line (HT29). Both systems showed that *Reg3* expression could be directly induced by the addition of hyaluronan fragments (6.8 kDa HA or LMWHA: Low molecular weight HA) (Fig. 2g–i). These findings were consistent with prior observations that hyaluronan fragments, acting as DAMPs in response to skin inflammation[19,20], can trigger other elements of the innate host defense system in the intestine[1,21–23]. Overall, these observations connect the presence of skin inflammation, subsequent release of hyaluronan fragments, and altered gene expression in the intestine.

### Skin injury changes the composition and behavior of the gut microbiome

Having observed the expression of *Muc2* and *Reg3* in the colon after skin injury, we next asked if skin wounding would cause a change in the composition of the intestinal microbiota. To address this, shotgun metagenomic DNA sequencing was performed on co-housed littermates with and without skin wounding ($n = 32$ in each group). A comparison cohort was provided oral vancomycin, a non-absorbable antibiotic that is limited to action within the intestine. This experimental group was used to compare changes in gut microbiota after skin wounding to the large changes in the gut microbiome that are

 

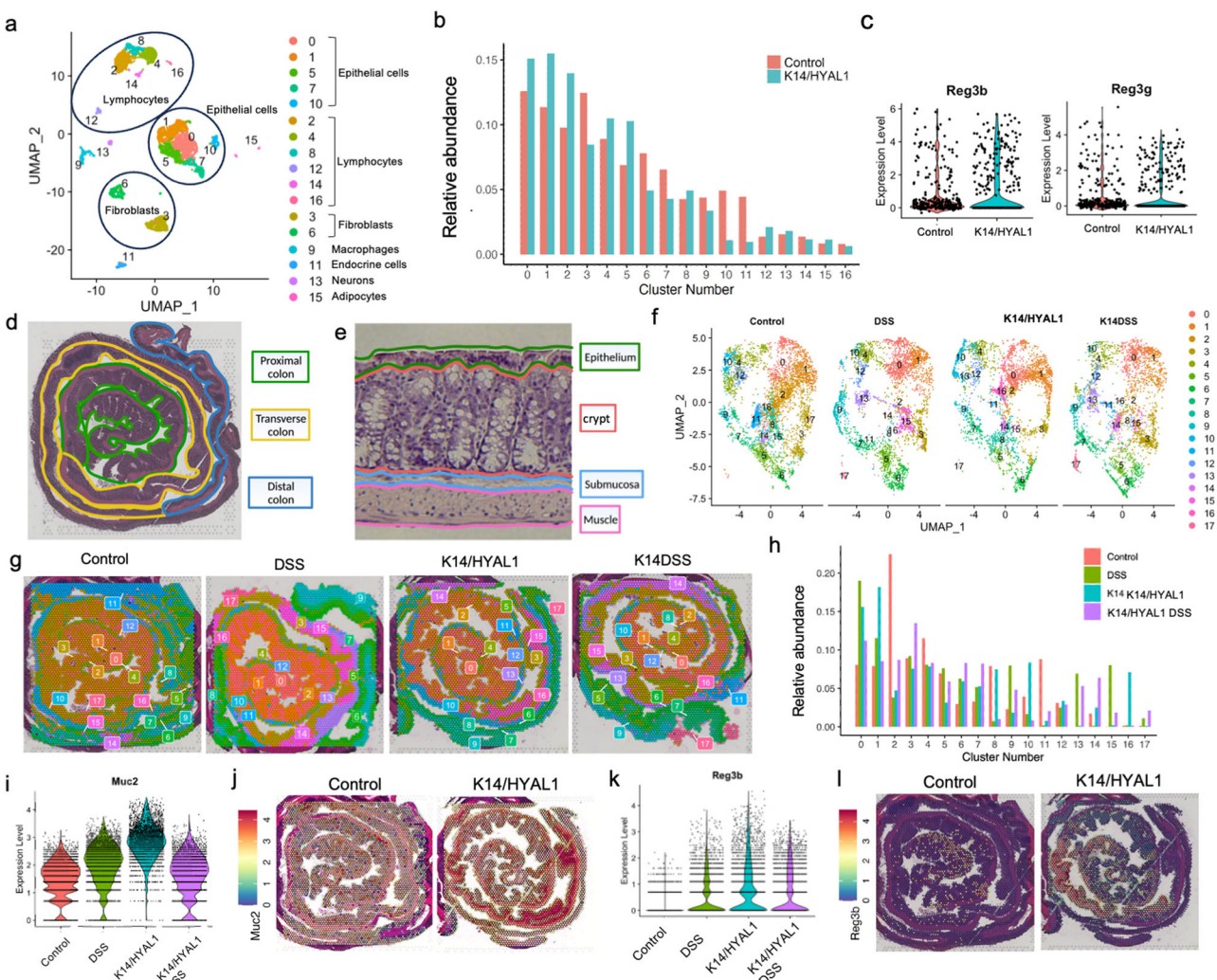

**Fig. 1 | Skin influences gene expression in the intestine and increases expression of *Muc2* and *Reg3*.** Single-cell RNA sequencing defines 17 cell clusters in the mouse colon. **a** UMAP plot. **b** Differential percent abundance in *K14/HYAL1* and control mice. **c** Violin plot of *Reg3* expression in cluster 10. **d, e** Spatial sequencing landmark slide with proximal, transverse, and distal colon delineated by color. **f** UMAP plot of spatial sequencing from Control, DSS, *K14/Hyal1*, and *K14/Hyal1* DSS mouse colon. Clusters are indicated by color and number. **g** The spatial representation of clusters in Control, DSS, *K14/Hyal1*, and *K14/Hyal1* DSS mouse colon. **h** Percentage abundance of each cluster in the sample. **i** Violin plot of *Muc2* expression. **j** Spatial plot of *Muc2* localization and abundance. **k** Violin plot of *Reg3b* expression. **l** Spatial plot of *Reg3b*.

induced by this broad-spectrum antibiotic. Although vancomycin induced the largest changes in gut microbiota, a significant difference was also observed in both Shannon alpha diversity and Robust Aitchison beta diversity between control mice and mice 2 days following a skin wound (Fig. 3a, b). The relative abundance of several bacterial species including *Lachnospiraceae bacterium A4* and *Akkermansia muciniphila* was increased in mice with skin wounds (Fig. 3c, d). Further functional analysis of the log-ratio of the top 20 changes in GO terms associated with bacteria showed that skin wounding was associated with the increased presence of genes in the gut microbiota that are related to the choline catabolic process and cobyrinic acid synthase activity, processes that are influence bacterial survival and the activity of opportunistic pathogens such as *Bacteroides thetaiotaomicron* (Fig. 3e). In addition to the increase in genes associated with pathogenic organisms, skin wounding also resulted in a decrease in genes that are considered beneficial to the intestine, such as propionate catabolic processes which contribute to suppress inflammatory response in the intestine[24]. Thus, skin injury resulted in a potentially negative effect on the gut microbiome as seen by a loss of beneficial organisms and a gain of pathogenic bacteria.

We next assessed bacterial viability and potential function of gut microbes from *K14/HYAL1* mice and mice with skin wounds. Stool samples from mice with skin wounds or *K14/HYAL1* mice had a decrease in total live bacteria and an altered cell morphology as assessed by FACS analysis when compared to cohoused littermates (Fig. 4a, b). A loss in absolute bacterial abundance was also shown by qPCR analysis for *16 S* rDNA (Fig. 4c). Primers specific for *A. muciniphila*, *L. gasseri*, and *B. thetaiotaomicron* also showed a decreased absolute abundance of these species in stool from mice with skin wounds (Supplementary Fig. 7a–c). This loss in viability of some bacteria in the colon was consistent with the observation of increased *Muc2* and *Reg3*.

Next, an assessment of tissue-associated bacteria was conducted of the transverse colon to determine if bacteria that survive in the colon following skin injury were distributed differently at the luminal interface. This analysis showed a large increase in Gram-negative bacteria within intestinal crypts and in the submucosal muscle layer of *K14/HYAL1* or skin wound mice (Fig. 4d, e and Supplementary Fig. 8a, b). This increase in bacteria within the tissue was confirmed by in-situ hybridization for *16 S* rRNA (Fig. 4f, g and Supplementary

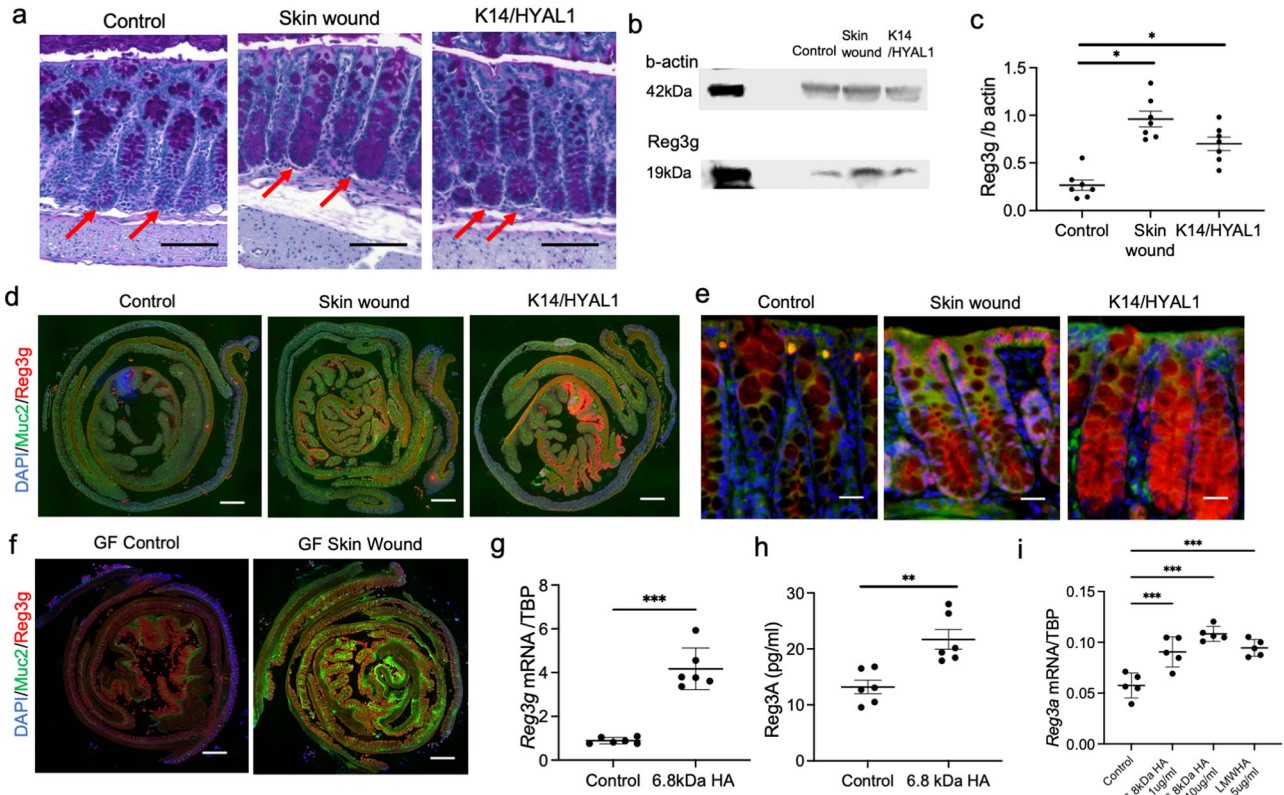

**Fig. 2 | Mucin and Reg3 protein expression increases in the intestine in response of skin wounding or exposure to hyaluronan. a** Representative image of periodic acid–Shiff (PAS) staining in the transverse colon. (scale bar: 100 micron, arrows point to Mucin staining in the intestinal crypt). **b** Western blotting of Reg3g extracted from transverse colon. **c** Quantification of staining intensity of Reg3g in the colon. ($n = 7$ independent biological replicates per group).
**d** Immunofluorescent staining of the colon. (Muc2: Green, Reg3g: Red DAPI: Blue. scale bar: 1000 micron). **e** High magnification of crypt structure. (Scale bar: 25 μm). **f** Immunofluorescent staining of the colon of germ-free mice with and without skin wounds. (Muc2: Green, Reg3g: Red DAPI: Blue). **g** mRNA expression level of *Reg3g* in colon treated ex-situ with hyaluronan 6.8kDA fragments ($n = 6$ independent

biological replicates per group). **h** Concentration of Reg3A in medium of colon epithelial cells (HT29) treated in culture with hyaluronan 6.8kDA fragments($n = 6$ biologically independent cells in each group). **i** Comparison of mRNA expression of *Reg3a* in colon epithelial cells following treatment with low molecular weight (LMW) hyaluronan and hyaluronan 6.8kDA fragments($n = 5$ biologically independent cells in each group). Statistical significance was determined using Student's unpaired two-sided *t*-test (**g** and **h**), ordinary one-way ANOVA and Tukey's multiple comparison two-sided test (**c** and **i**). Error bars indicate mean ± SD; * $P < 0.05$, ** $P < 0.01$, *** $P < 0.001$. Each experiment was repeated at least 3 times. Source data are provided as a Source Data file.

Fig. 8c). The increase in bacteria within the tissue was not due to an increase in epithelial permeability in the colon as no increase was detected by FITC-dextran penetration (Fig. 4h). These findings suggest that although there was a loss in total bacteria after skin injury, potentially due to the increase in mucin and *Reg3*, surviving organisms have an increased capacity to penetrate the mucus layer and penetrate the colonic epithelium.

## Changes to the gut microbiota after skin injury increases susceptibility to DSS

Since prior studies had shown skin wounding (and *K14/HYAL1* mice) resulted in increased susceptibility to colitis to DSS, we next sought to determine if the alterations in the intestinal microbiome might explain this phenomenon. To investigate if gut bacteria were involved in the susceptibility to DSS, *K14/HYAL1* mice and mice after skin wounding were treated with oral vancomycin. Alternatively, germ-free mice were also tested after skin wounding (Supplementary Fig. 9a, b). *K14/HYAL1* mice challenged with DSS showed increased expression of *TNF* in the colon compared to controls, and this increase was eliminated by treatment with vancomycin (Fig. 5a). Similarly, vancomycin also eliminated the increase in intestinal *TNF* expression in mice with skin wounds who were exposed to DSS (Fig. 5b). Consistent with the action of TNF to induce cell death, an increase in apoptotic cells was observed by TUNEL staining in mice with DSS and skin wound or *K14/HYAL1*

compared to control mice (Supplementary Fig. 10a, b). Inhibition of TNF abrogated the increased inflammation after DSS as assessed by FACS for neutrophils and macrophages (Supplementary Fig. 10c–e). However, although these observations support the important role of TNF in enabling skin-specific interventions to exacerbate colitis, this inflammatory response was dependent on the presence of bacteria in the gut as skin wounds did not increase *TNF* in the colon of germ-free mice (Fig. 5c) and mortality rate, body weight loss, tissue histology, and infiltration of neutrophils and macrophages was rescued after treatment with vancomycin (Fig. 5d–h and Supplementary Fig. 9c–h). These data demonstrated that bacteria in the intestine are important for the increased disease severity following DSS.

Finally, to demonstrate that the fecal microbiome from mice with skin wounds could cause enhanced susceptibility of the intestine to DSS, we performed fecal microbiome transplantation (FMT) from mice with or without skin wounds (Fig. 5i). DSS administration to germ-free mice that received FMT from mice with skin wounds showed higher intestinal inflammation compared to germ-free mice that received FMT from co-housed littermates without skin wounds. Recipients of FMT from wounded mice had elevated expression of *IL6* (Fig. 5j), shortened colon length (Fig. 5k), histological evidence of increased tissue damage (Fig. 5l), and increased inflammatory cells within the colon (Fig. 5m) compared to mice that received FMT from control mice. Germ-free mice that received FMT from mice with skin wounds

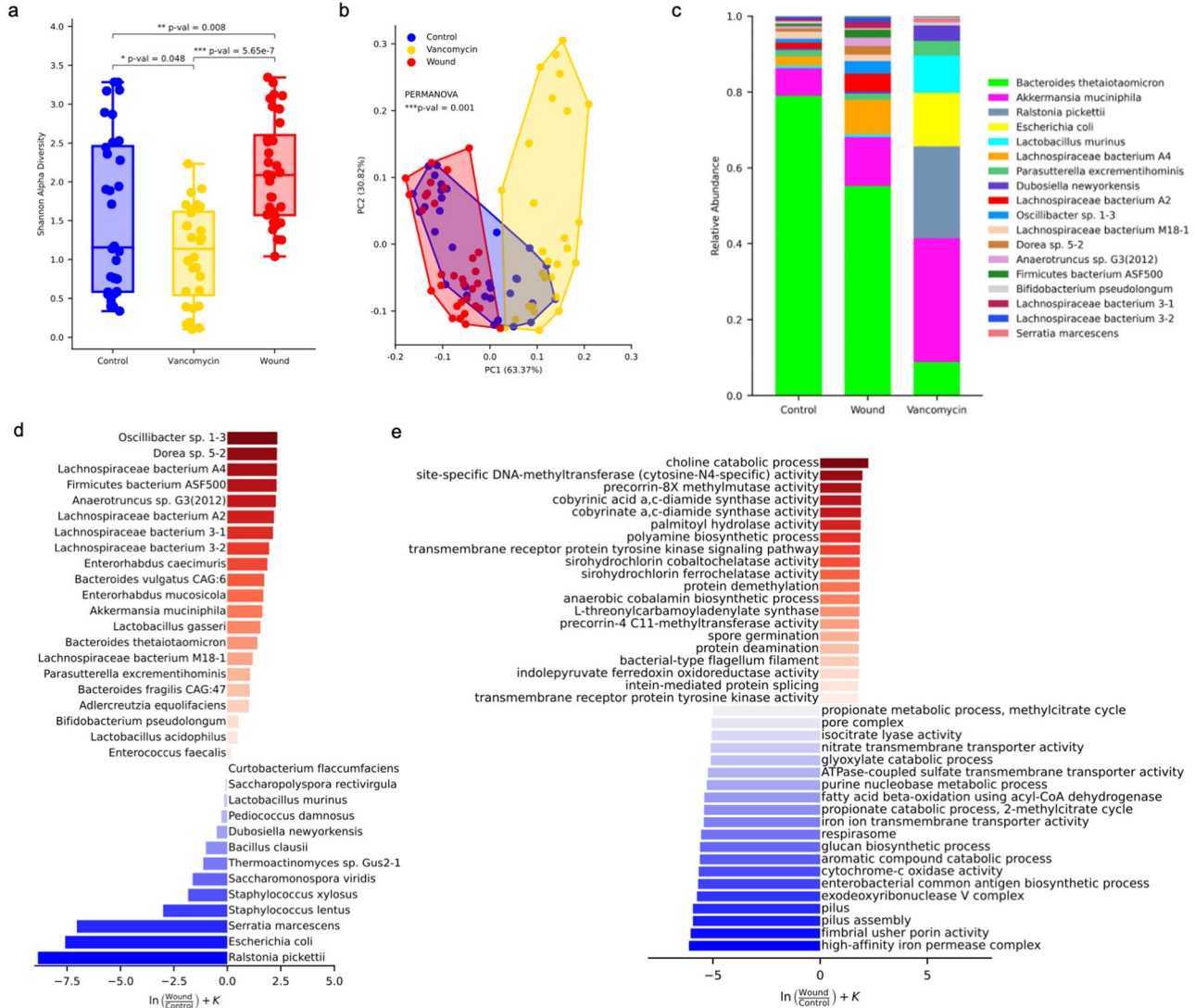

**Fig. 3 | Skin injury changes the composition of the gut microbiome. a** Alpha diversity analysis by the Shannon index and significance testing between groups by the Mann Whitney *U* test. Box plots represents mouse type, with the center line indicating the median, the box bounds representing the 25 and 75 percentiles, and the whiskers extending to minima and maxima, 1.5 times the interquartile range from the 25th and 75th percentiles, respectively. *N* = 32 control, 28 vancomycin, and 32 wound. **b** Beta diversity analysis using robust Aitchison PCA and significance testing between groups by PERMANOVA. **c** Relative abundance of top 34 bacterial species (top 18 shown in the legend). **d** Metagenomic differential abundance by Songbird of bacterial species plotted by their log-ratio associated with wound (red) and control (blue). **e** GO terminology functional differential abundance by Songbird.

also showed increased *Muc2* and *Reg3g* compared to FMT from control mice without wounds (Supplementary Fig. 11a–c).

## Discussion

Our results show that localized injury to the skin will alter intestinal antimicrobial defenses and change the gut microbiome. This change in bacteria within the gut increases susceptibility to DSS colitis. The HA digestion model was used to recapitulate the local release of hyaluronan fragments from the dermis that occurs with wounding or other forms of skin inflammation such as psoriasis[25], and control for confounding effects that also occur with skin inflammation. These findings provide an unexpected explanation for the association between skin and intestinal diseases in humans[8,26,27]. Although prior studies have observed dysbiosis in the gut microbiome of individuals with inflammatory skin disease, it had been assumed that microbes in the gut were influencing the skin[28–30]. Our observations suggest an alternative explanation; skin inflammation will change the composition and functions of the gut microbiome. Given the expanding evidence that the intestinal microbiome is associated with changes in the function of

other organ systems[12,13,31], our results also suggest that the skin can also affect other organ systems such as the lung or brain. Therefore, we demonstrate the interconnectivity of epithelial barrier tissues and illustrate how host-microbe interactions and the microbiome of one tissue can be influenced by a distal epithelial tissue.

Communication between organ systems is a complex network that likely involves several factors. Our data show that the release of hyaluronan fragments from the dermis after skin injury is relevant to this process. The strength of the *K14/HYAL1* is that it addressed potential confounding variables associated with skin injury since these mice do not have an increase in local skin inflammation, abnormalities in skin development, or leukocyte migration from the skin to the gut that could confound interpretation of wounding as specific to the skin. This model therefore served to validate that a skin-specific intervention was promoting the changes observed in the gut[15]. Although a major intestinal phenotype was previously seen in mice with skin wounds or *K14/HYAL1* after a challenge with DSS, it remained unclear how these skin changes caused the increase disease in the intestine. Current findings now link these processes by showing that changes in

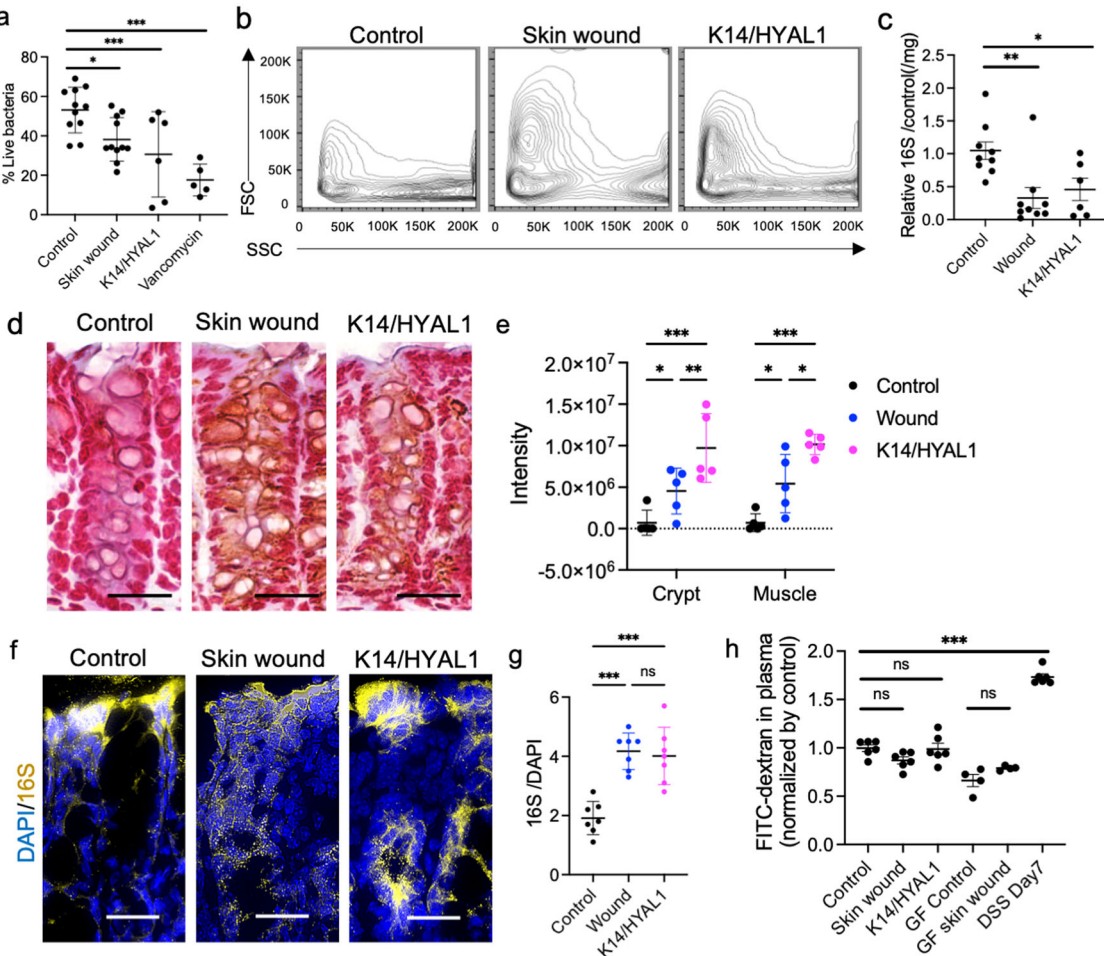

**Fig. 4 | Skin injury or dermal hyaluronidase expression alters bacterial survival and epithelial penetration in the intestine. a** Proportion of live bacteria in feces as measured by Flow cytometer (control, skin wound: n = 11, *K14/HYAL1*: n = 6, Vancomycin: n = 5 independent biological replicates per group). **b** Bacterial morphology as measured by flow cytometer. **c** qPCR measurement of relative abundance of 16 S rDNA per mg feces (control, skin wound: n = 9, *K14/HYAL1*: n = 6 independent biological replicates per group). **d–e**. Gram staining of bacteria in the transverse colon at (**d**). Low magnification (scale bar: 50 micron) and **e** Intensity of gram staining in crypt or muscle layers from control, skin wound, *K14/HYAL1*. (n = 5

independent biological replicates per group). **f** In situ hybridization assay of bacterial 16 S rDNA in the colon. (scale bar: 50 micron, arrows). **g** Intensity of 16 S signal normalized by DAPI. (n = 7 independent biological replicates per group). **h** Concentration of FITC-labeled dextran sulfate entering the plasma after oral gavage (SPF: n = 6, GF: n = 4 independent biological replicates per group). Statistical significance was determined using ordinary one-way ANOVA and Tukey's multiple comparison two-sided test. Error bars indicate mean ± SD; * $P < 0.05$, ** $P < 0.01$, *** $P < 0.001$. Each experiment was repeated at least three times.

the gut microbiome that occur due to HA release from the skin are responsible for increased colitis after DSS.

HA fragments are an important DAMP that activates local defense responses in tissues such as skin, gut, and lung, and these responses are triggered by several mechanisms including the receptors CD44 and TLR4[32–34]. Since *K14/HYAL1* mice recapitulated many of the gut responses observed following skin wounding, this suggests that HA digestion is one mechanism by which skin wounding, and potentially other forms of skin inflammation including psoriasis, can communicate with the colon. We show psoriasis is associated with increased digestion of HA in the dermis and our data from experiments done in vitro directly demonstrate that HA fragments enhanced Reg3 production in mouse colon and in cultured human colon epithelial cells. This observation is also consistent with prior findings that HA fragments promoted Reg3 production in the colon through TLR4[19]. In addition, HA fragments were previously shown to also enhance the capacity of submucosal fibroblasts to undergo adipogenesis[15]. It remains to be studied if the fibroblast response to HA and subsequent enhanced adipogenesis in the colon is part of the system that drives intestinal dysbiosis.

Interestingly, data from spatial sequencing of the proximal and transverse colon, which arise embryologically from the midgut, showed different transcriptional profiles and a greater response to HA fragments than other sites in the intestine. The change in *Reg3* and *Muc2* expression indicates that either gut bacteria were changed by the induction of host defense gene expression or that the change in microbes caused the change in expression of the host defense genes. However, the latter cannot be the sole factor as skin wounding was able to induce *Reg3* and *Muc2* expression in germ-free mice. Furthermore, although no change in resident immunocytes was detected in this study after skin wounds, or in prior evaluation of trafficking of cells from skin to gut of *K14/HYAL1* mice[15], other DAMPs and immunologically active molecules may also participate in the communication from the skin to the intestine. For example, transcriptional analysis detected changes in ISG15 Ubiquitin Like Modifier (*Isg15*), LY6/PLAUR Domain Containing 8 (*Lypd8*), and Integrin beta-6 (*Itgb6*), genes that have also been reported to influence host defense. Furthermore, since skin wounding of germ-free mice could induce *Muc2* and *Reg3g* expression in the colon, and FMT from mice with skin wounds could also induce *Muc2* and *Reg3g* in germ-free mice without skin injury, it is likely that

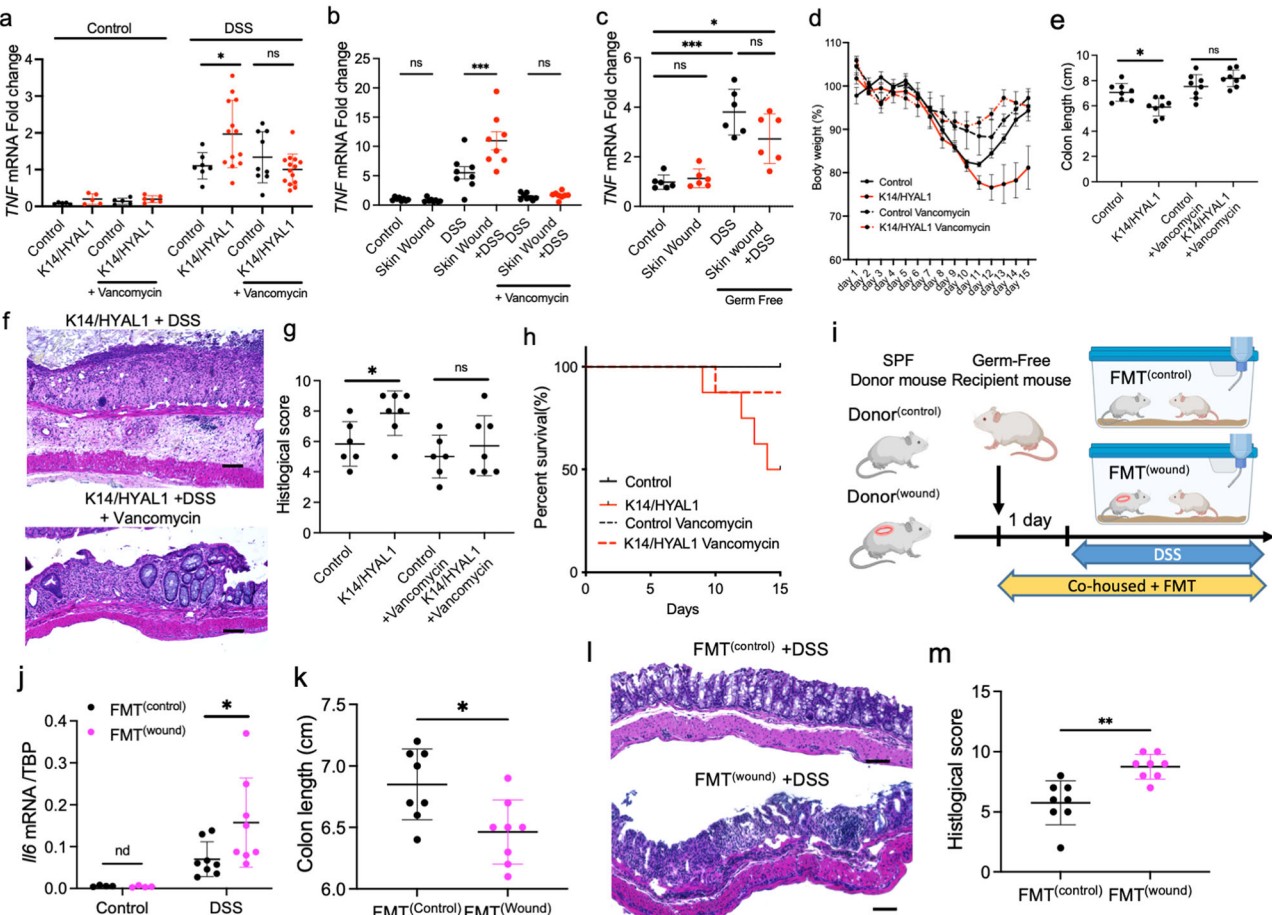

**Fig. 5 | The capacity of the skin to promote DSS colitis is dependent on bacteria in the gut. a** TNF mRNA expression in the colon of mice after DSS challenge to control or *K14/HYAL1* mice with or without pretreatment by oral vancomycin (50 mg/kg) (No treatment: *n* = 5, DSS treatment: control *n* = 8, *K14/HYAL1* n = 13). **b** TNF mRNA expression in the colon of mice after DSS challenge following skin wounding with or without pretreatment by oral vancomycin (*n* = 8). **c** TNF mRNA expression in the colon of germ-free mice after DSS challenge following skin wounding (*n* = 6). **d** Percent change in body weight normalized to weight at day 0. **e** Colon length at the 14 days after beginning of DSS treatment (*n* = 8 independent biological replicates per group). **f, g** Histological images of the distal colon from control and K14/HYAL/1 mice treated with or without vancomycin and Disease activity index (*n* = 6). **h** Survival rate of control of *K14/HYAL1* mice over time after administration of DSS (*n* = 8. Scale bar: 50 μm). **i** Schematic of FMT experiments created with BioRender.com. **j** *Il6* mRNA expression in the colon of mice following FMT from mice with or without skin wounds and challenge by DSS (*n* = 8). **k** Colon length of mice following FMT and 5 days after the beginning of DSS treatment (*n* = 8). **l, m** Histological images of the distal colon of mice following FMT and 5 days after the beginning of DSS treatment and Disease activity index (*n* = 8. Scale bar: 50 μm). Statistical significance was determined using Student's unpaired two-sided *t* test (**k**), ordinary one-way ANOVA and Tukey's multiple comparison two-sided tests (**a–c**, **e**, **g**, **j**, **k** and **m**) and ordinary two-way ANOVA and Sidak's multiple comparisons two-sided test (**d**). Error bars indicate mean ± SD; * *P* < 0.05, ** *P* < 0.01, *** *P* < 0.001. Each experiment was repeated at least three times. Source data are provided as a Source Data file.

signals both from the skin and the intestinal microbes alter intestinal host defense.

Both mucin and Reg3g participate in enforcing spatial separation of the microbiome from the host epithelium[23,35]. Therefore, the increased bacterial penetration of the intestinal epithelium in response to skin wounding was unexpected. However, excess production of AMPs could lead to a loss of bacterial species that regulate the behavior of other microbes, ultimately contributing to the disruption of homeostasis in the intestine. Similarly, increases in *Muc2* may not directly result in increased mucus protection, and depends on the composition of other mucus components. Microbial penetration past the mucus layer that we observed may have been due to higher expression of genes associated with the choline catabolic process and cobyrinic acid synthase activity processes, and these are associated with the activity of opportunistic pathogens such as *Bacteroides thetaiotaomicron*. Furthermore, these more virulent microbes may have been selected in the colon as a consequence of the skin-altering intestinal antimicrobial defense, and the surviving bacteria may have led to increased disease. This is supported by the observation that a lesser abundance of total bacteria was measured in the feces after skin wounding, but the strains surviving in the intestine after skin injury had genes associated with enhanced capacity to resist barrier defenses and penetrate the epithelium. The relative roles of *Muc2* and *Reg3g* in this process, and the specific characteristics of the bacteria that penetrate the intestinal epithelium, require further study.

Alterations in the composition of the microbiome have been associated with variety of disease conditions in several organs[11,31,36,37]. Since, FMT from mice with skin wounds to germ-free mice increased the recipient's susceptibility to DSS we propose that the increase in susceptibility to DSS after skin injury was due to this change in the intestinal microbiome. Although metagenomic analysis showed skin wounding increased the relative abundance of species such as *Akkermansia muciniphila* and *Lactobacillus gasseri* that are often considered to be beneficial[38], the quantity and viability of fecal bacteria was lower after wounding. Furthermore, GO term analysis showed a decrease in the abundance of genes associated with a health benefit such as the propionate metabolic process. We speculate that the observations of increased bacteria within the crypts and deeper tissue reflect a shift to

more invasive organisms and is part responsible for subsequent increased disease after DSS challenge.

Overall, our results show that the skin, as the barrier tissue with the most direct exposure to the environment[36,39,40], can cause dysbiosis of the gut and influence susceptibility to disease. Such an observation supports better efforts to care for the skin, and therapeutic interventions that consider the microbiome and health of other epithelial tissues when treating IBD.

## Methods

Study approval: All animal experiments were approved by the University of California San Diego, Institutional Animal Care Committee (IACUC) S09074.

### Animals and animal care

For all animal studies, animals were randomly selected without formal pre-randomization, and quantitative measurements were done without the opportunity for bias.

Transgenic mice for conditional overexpression of human hyaluronidase-1 (in C57BL/6 background) were generated in our laboratory as described previously by combining a constitutive promoter and a loxP-floxed GFP reporter upstream of hyaluronidase-1 (Hyal1)[25]. Wildtype mice, germ-free, and K14-cre transgenic mice were in C57BL/6 background and originally obtained from The Jackson Laboratory. K14-cre transgenic mice were bred with Hyal1 mice for the generation of K14-cre Hyal1 mice. Control mice used for comparison to K14-cre Hyal1 mice were littermates that were negative for Cre and heterozygous for Hyal1. Germ-free mice were bred and housed in flexible film isolators until 6–8 weeks of age and transferred to SPF facility at UCSD. 8-12-week-old male and female mice were utilized and samples comprised an equal number of male and female mice to ensure balanced representation across sexes. These facilities maintain a 12/12 dark/light cycle, 65–75 °F with 40–60% humidity. All mice are euthanized by $CO_2$.

### Mouse model of DSS colitis

Wildtype (WT) mice (C57BL/6 mice) or K14-cre Hyal1 mice (K14/HYAL1) were provided 2.5% DSS (Dextran sulfate sodium salt, colitis grade (36,000 –50,000), MP biomedicals) in their drinking water for 7 days and body weight was measured every day. Mice were sacrificed on Day 14 after DSS administration (no treatment: $n = 5$, DSS treatment: control $n = 8$, K14/HYAL1 $n = 13$). Mice that lost more than 20% of their original body weight were euthanatized. Skin wound or control wildtype mice in SPF ($n = 3$), or germ-free conditions ($n = 6$), or fecal microbiome transplantation mice ($n = 8$) were provided 2.5% DSS in their drinking water for 5 days and sacrificed. Histological severity was evaluated by the disease activity index based on these criteria.

(crypt architecture (normal, 0 - severe crypt distortion with loss of entire crypts, 3),

degree of inflammatory cell infiltration (normal, 0 - dense inflammatory infiltrate, 3),

muscle thickening (base of crypt sits on the muscularis mucosae, 0 - marked muscle thickening present, 3),

goblet cell depletion (absent, 0- present, 1) and crypt abscess (absent, 0- present, 1).)[41]. For anti-TNF treatment, 50 μg of anti-mouse TNF (Clone:XT3.11, Bio X Cell, NH, USA) was injected intraperitoneally every other day from 2 days prior to providing DSS ($n = 5$).

### Mouse model of skin wounding

Skin wound experiments were done as described[42]. In brief, the backs of sex-matched and age-matched (8–12 weeks) adult wildtype mice were given a 1.5 cm full thickness incisional wound under anesthesia by isoflurane. Wounds were left open after injury and not closed by suture. Forty-eight hours later, mice were provided 2.5% DSS in their

drinking water for 5 days. Surgical sterile techniques were used in handling mice within the germ-free facility.

### Mouse colon ex vivo model

The whole mouse colon was resected from C57BL/6 mice and initially flushed with 5 ml of sterile DMEM culture medium supplemented with 10% (v/v) FBS and 1% antibiotic solution (Antibiotic-Antimycotic (100X)). During treatment, the excised and flushed colon was tied with sterile suture on both ends after filling with DMEM culture medium or DMEM with 5 ug/ml HA($n = 6$). Colons were incubated in a humidified incubator at 37 °C with 5% CO2 for 15 m prior to assessment of the response to HA. HA 6.8 kDa fragments were obtained from Seikagaku Corporation (Tokyo, Japan).

### Cell culture

HT-29 human CRC cells were purchased from the American Type Culture Collection. HT-29 cells were grown in T75 flasks using DMEM culture medium supplemented with 10% (v/v) FBS and 1% antibiotic solution (Antibiotic–Antimycotic (100x)), respectively. The cells were maintained in a humidified incubator at 37 °C in 5% CO2. HA forms of 6.8 kDa were provided by Seikagaku Corporation (Tokyo, Japan). Low molecular weight HA (LMWHA) was digested from HA from streptococcus equi (#906327, Sigma Aldrich) by incubation with Hyaluronidase (#H5306, Sigma Aldrich) at 37 °C for 24 h and 65 °C for 20 min.

### Intestinal permeability assay

Mice were gavaged with FITC-Dextran (4 kDa, Sigma Aldrich) as previously described 3 h prior to fluorometric analysis of FITC fluorescence in plasma[43]. (n: WT = 6, Skin wound = 6, K14/HYAL1 = 6, GF control = 4, GF skin wound = 4, DSS Day7 = 6)

### Bacterial culture and count

Fecal samples were shaken and resolved in the sterilized PBS with 10% glycerol. (1 g feces are resolved in 2 ml solution). Bacteria were stained by LIVE/DEAD BacLight Bacterial Viability Kit (Thermo Fisher) and counted by the flow cytometer according to the manufacturer's protocol by the BD FACSCanto RUO machine and analyzed by FlowJo V10 software. (n: Control = 11, Skin wound = 11, K14/HYAL1 = 6, Vancomycin = 5)

### Fecal microbiome transplant (FMT)

Mouse feces are collected from the control mice or skin wound mice and stored in the sterilized PBS with 10% glycerol. (1 g feces are resolved in 2 ml solution) For FMT preparation, samples are mixed and eliminated the particles by centrifuge (500 x g for 5 min) and aliquots are stored at −80 °C. Aliquots are diluted five times and gavage 200 ul/mouse for 6 days until the end of the DSS challenge. Gavage was performed every day during the experiment. ($n = 8$).

### Human skin sample collection

Fresh adult human skin biopsies, from the back of healthy donors (age 18–50), were collected from the Dermatology Clinic, University of California, San Diego (UCSD) and from the Dermatology Clinic, Sample acquisitions were approved and regulated by the UCSD Institutional Review Board (IRB; reference number 140144). Biopsies were immediately embedded in the Tissue-Tek optimal cutting temperature (OCT) compound for sectioning and staining. The informed consent was obtained from all participants before skin biopsies. Upon collection, these samples were directly OCT-embedded for immunofluorescent analyses.

### Histology and immunohistochemistry (IHC)

Tissue biopsies were directly embedded in the OCT compound or fixed with Carnoy's fluid. Paraffin-embedded tissues were used for hematoxylin/eosin (H&E) staining, and frozen sections were fixed in 4%

PFA for 20 min to immunofluorescence staining. For Gram staining, tissue is fixed with Carnoy's fluid and embedded in OCT compound, sectioned by 4um and stained with Gram stain (Remel, Lenexa, KS). For Periodic acid-Schiff(PAS) staining, tissue is fixed with Carnoy's fluid and embedded in Paraffin, and stained with PAS. For TUNEL staining, TUNEL Andy Fluo 594 Apoptosis Detection Kit (#A051, abpbio) is used. For IHC, fixed and permeabilized frozen tissue sections were blocked with Image-iT FX reagent (Invitrogen) before incubating with HABP (#385911, EMD Millipore), Muc2 (#PIMA512345, Fisher) or Reg3g (#PA5-50450, Thermo Fisher) in 1:100 dilution followed by appropriate 488- or 568-coupled secondary antibodies. Nuclei were counterstained with DAPI. All images were taken with an Olympus BX41 microscope (widefield) or Zeiss LSM510 confocal microscope as indicated.

### In situ hybridization
Fresh frozen colon tissue sections were obtained and fixed in Carnoy's fluid overnight. Sections were stained for 16 S using the RNAscope Fluorescent Multiplex Assay (Advanced Cell Diagnostics Bio) according to the manufacturer's protocol with a predesigned probes (Cat # 422521).

### IHC quantification
Images from PAS staining, gram staining, and 16 S signals in colonic tissue sections were quantified using the color thresholding tool in Fiji (ImageJ). Each condition is as follows (For PAS staining: Hue threshold: 188–216, Saturation threshold: 120–255, Brightness threshold: 0–122. For gram staining: Hue threshold: 10–41, Saturation threshold: 120–255, Brightness threshold: 143–255).

### ELISA
Cell culture supernatants were isolated, and cellular debris was removed by centrifugation at 600×g for 5 min. The supernatant was frozen at −80 C until use for analysis. Reg3A ELISA was performed with sandwich ELISA kits (Cat# DY5940-05, R&D Systems) according to the manufacturer's instructions. ELISA results were quantified on a Spectramax Absorbance reader (Molecular Biosystems).

### Western blotting
Mouse colon tissues were homogenized lysed in RIPA buffer (Thermo Fisher). After centrifugation, cell lysates were subjected to SDS gel electrophoresis and transferred onto polyvinylidene difluoride membranes (IPVH 00010, Millipore). The membranes were analyzed by immunoblotting with the indicated antibodies.

### Tissue processing for single cell RNA sequencing
Tissue samples from 3 mice in each group were minced with a razor blade into 1 cm fragments, suspended in enzymatic digestion buffer collagenase and DNase I as previously described[44], incubated with frequent agitation at 37 °C for 30 min, and triturated briefly with a 5 ml pipet. Cells in a single-cell suspension were then passed through a 100-micron mesh filter, centrifuged, and stained with a live/dead stain for FACS sorting for cells of fibroblast lineage. In total 20,000 sorted cells were loaded on the 10X Genomics Chromium system. Library construction protocol: Single-cell suspensions were loaded onto the 10X Genomics Chromium Controller instrument to generate single-cell GEMs. GEM-RT and library construction were performed following the 10X Genomics Protocol. Library fragment size distributions were determined using an Agilent Bioanalyzer High Sensitivity chip, and library DNA concentrations were determined using a Qubit 2.0 Fluorometer (Invitrogen). Libraries were sequenced using an Illumina NovaSeq.

### Spatial transcriptomics
Colonic tissues from untreated and DSS-treated mice from WT and K14/HYAL1 mice were cleaned from adipose tissue and cut longitudinally; the luminal content was removed by washing it in cold phosphate-buffered saline (PBS). Starting from the most proximal portion (i.e., Cecum) and with the luminal side facing upward, the colon was rolled resulting in a roll with the proximal colon in the center and the distal colon in the outer portion of the roll. The roll was placed in a histology plastic cassette and snap-frozen for 1 min in a bath of liquid nitrogen-cooled isopentane. The frozen tissue was then embedded in an Optimal Cutting Temperature compound (OCT, Sakura Tissue-TEK) on dry ice and stored at −80 °C. OCT blocks were cut with a pre-cooled cryostat at 10 micron thickness, and sections were transferred to fit the 6.5 mm$^2$ oligo-barcoded capture areas on the Visium 10x genomics slide. Before performing the complete protocol, Visium Spatial Tissue Optimization (10x Genomics) was performed according to the manufacturer's instructions, and the fluorescent footprint was imaged using a Metafer Slide Scanning Platform (Metasystems). 9 min was selected as the optimal permeabilization time. The experimental slide with colonic tissue was fixed and stained with hematoxylin and eosin (H&E) and imaged using a Keyence BZX-700 Fluorescent Microscopy (Keyence) at 2× magnification. Sequence libraries were then processed according to the manufacturer's instructions (10x Genomics, Visium Spatial Transcriptomic).

### Data analysis
For mouse colon, the 10X Genomics Cell Ranger version 7.0.1 and Space Ranger version 2.0.1 software pipeline with default parameters were used to perform sample demultiplexing, barcode processing, alignment to the mm10 reference genome, and single-cell gene counting. Data were further filtered, processed, and analyzed using the Seurat R toolkit version 4.0.6[45,46]. Filtering of initial data involved selecting cells with >200 and <3000 features and <15% mitochondrial genes. For single-cell sequencing, data were combined by IntegrateData(). The data were normalized using the SCTransform() function with parameters normalization. "Method" = 'SCT'. Prior to integrating the data, PrepSCTIntegration(), Find IntegrationAnchors(). Data were scaled with the ScaleData(). Principal components were calculated from these variable genes using the function RunPCA(). Nonlinear dimensionality reduction and visualization were performed with UMAP[47] using the RunUMAP()function. Clusters were identified using the FindNeighbors()function using the significant PCs then the FindClusters() function with the parameter resolution = 0.5. Marker genes for clusters and between samples were determined using the FindAllMarkers()function with parameters min.pct = 0.25 and thresh.use = 0.25. Gene ontology analysis was performed on marker genes using the 'clusterProfiler' R package with default parameters[48]. The Seurat object was transformed into a SingleCellExperiment object. Marker genes were identified using the FindAllMarkers() function with parameters test.use = "LR", latent.-vars = "Exp", min.pct = 0.25, and logfc.threshold = 0.4054651 (corresponding to 1.5-fold change).

For Spatial sequencing, data were combined by merge(). Principal components were calculated from these variable genes using the function RunPCA(). Nonlinear dimensionality reduction and visualization were performed with UMAP[47] using the RunUMAP()function. The data were normalized using the SCTransform()function with parameters normalization.method = 'SCT'. Clusters were identified using the FindNeighbors()function using the significant PCs and then the FindClusters() function with the parameter resolution = 0.5. Marker genes for clusters and between samples were determined using the FindAllMarkers()function with parameters min.pct = 0.25 and thresh.use = 0.25. Gene ontology analysis was performed on marker genes using the 'clusterProfiler' R package with default parameters[48]. The Seurat object was transformed into a SingleCellExperiment object. Marker genes were identified using the FindAllMarkers() function with parameters test.use = "LR", latent.vars = "Exp", min.pct = 0.25, and logfc.threshold = 0.4054651 (corresponding to 1.5-fold change).

## Flow cytometry analyses

Colon collected from control or DSS-treated mice was cut into small pieces then digested with 2.5 mg/mL Collagenase D and 30 ng/mL DNAse1 for 40 min at 37 °C then filtered through a 70 μm filter to generate single cell suspension for FACS analyses. Cells were then stained with Fixable Viability Dye eFluor 506 (eBioscience, 65-0866-14), blocked with anti-mouse CD16/32 (eBioscience, 14016185), followed by staining with antibody cocktails for preadipocytes or immune cells. The antibody cocktail for immune cells includes FITC -CD45 (BioLegend, 103107), PECy7-CD11b (BioLegend, 101216), FITC-Ly6G (eBioscience, 11593182), PE-F4/80 (eBioscience,12480182), APC-CD11C (BioLegend, 117310), AF700-MHCII (eBioscience, 56532182), and APC-eFluro-CD4 (eBioscience, 47-0042-80) PE -CD19 (BioLegend, 115507). All antibodies were used at a final dilution of 1 to 100. FACS analyses for surface expression of immune cell markers were performed by the BD FACS-Canto RUO machine and analyzed by FlowJo V10 software.

## Reverse transcription–quantitative PCR (RT–qPCR) analyses

RTqPCR was used to determine the mRNA abundance. The total cellular RNA was extracted using the PureLink RNA Mini Kit (Life Technologies Corporation). In all,100 ng of mRNA was reverse transcribed to cDNA using Verso cDNA Synthesis Kit (Thermo Fisher Scientific Inc). Quantitative, real-time PCR was performed on the CFX96 real time system (Biorad) using a predeveloped Taqman gene expression assay (Applied Biosystems) or SYBR Green Mix (Bimake, Houston, TX). The housekeeping gene *Tbp* (TATA-binding box protein) was used to normalize gene expression in samples. Specific primer sequences are shown in Supplementary Table 1.

## Microbiome analysis

Co-housed C57/Bl6 mouse fecal samples were collected from control unwounded mice, skin wound mice, and vancomycin treated mice (*n* = 32, two control and three skin wound mice or three vancomycin treated mice co-housed in each cage). Bacterial DNA for shotgun metagenomic sequencing was extracted by the QIAamp DNA Stool Mini Kit (QIAGEN). Bacterial DNA for qPCR was extracted by the HostZERO Microbial DNA Kit (Zymo Research Corporation). Library construction was performed as previously described[49]. Shotgun metagenomic sequencing was done at the UC San Diego IGM Genomics Center utilizing an Illumina NovaSeq 6000.

We processed the metagenomics data using the Woltka[50] pipeline through Qiita[51]. This pipeline includes adapter and host filtering with qp-fastp-minimap2, sequence alignment against the Web of Life[52] (WoL) database using Bowtie2[53], and resultant taxonomic and functional classification with Woltka. The per-genome BIOM table[54]. was used to perform analysis on the observed operational genomic units (OGUs). This table was then filtered using Zebra filter[53]. based on genome coverage with a default 10% minimum coverage threshold.

Community (alpha and beta) analyses were performed in Python. For alpha diversity analysis, the taxonomic feature table was rarefied to the lowest sequencing depth and calculated using the Shannon Index via Python's scikit-bio package for alpha diversity. A Mann–Whitney U test was used through SciPy[55]. to determine the statistical significance of alpha diversity differences. For beta diversity, we used compositionally aware and phylogenetically informed robust principal-component analysis (Phylo-RPCA)[56]. on the unrarefied table using the WoL phylogenetic tree. PERMANOVA calculation was performed through scikit-bio to calculate statistical significance.

Next, we performed differential abundance analysis on the unrarefied table using Songbird[57]. We used 1000 epochs and a differential prior of 1.0 to determine the log-fold changes of taxa associated with mouse type (control vs. wound). A pseudocount of 1 was added to account for zeroes. Statistical testing of OGU log-ratios was performed using the Mann-Whitney U test from SciPy.

We used the per-gene BIOM table from Woltka to perform functional analysis. First, we used the collapse function in Woltka to group features into GO terms. We used the same parameters as outlined above to run Songbird on the GO table. We used the top and bottom 20 GO terms associated with wound status to compute log-ratios per sample.

Processing of tables was performed using Pandas and NumPy in Python. All plots were generated using Matplotlib and Seaborn.

## Statistics

Experiments were repeated at least three times with similar results. Statistical significance was determined using Student's unpaired two-tailed *t*-test, ordinary one-way ANOVA and Tukey's multiple comparison two-sided test, or ordinary two-way ANOVA and Sidak's multiple comparisons two-sided test as indicated in the legend (*$P < 0.05$, **$P < 0.01$, ***$P < 0.001$). For microbiome analyses, statistical significance was determined using non-parametric tests including the Mann-Whitney U test and PERMANOVA.

## Reporting summary

Further information on research design is available in the Nature Portfolio Reporting Summary linked to this article.

## Data availability

Single cell sequencing data generated in this study have been deposited in the GEO database under accession code (GSE227836). Spatial sequencing data have been deposited in GEO under accession code (GSM7109548). Microbiome data generated in this study is available at BioProject ID: PRJNA1003965. All other data are available in the article and its Supplementary files or from the corresponding author upon request. Source data are provided in this paper.

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

## Acknowledgements

Funding was obtained from National Institutes of Health R01DK128787 (R.L.G., N.S., R.K.), and R37AI052453 and R01AR076082 (R.L.G.) All the Sequencing was performed at the Institute for Genomic Medicine at

UCSD. Figure 5i and Supplementary Fig. 9a, b were created with BioRender.com.

## Author contributions

T.D. designed and performed experiments, analyzed data and wrote the paper; S.B., H.S., E.A.B., Y.N., A.O., and F.L. performed experiments and analyzed data; Y.C., G.R., D.H., and K.J.C. analyzed data. R.K. and N.H.S. supervised; R.L.G. designed experiments, supervised, analyzed data, and wrote the paper.

## Competing interests

Ethics and Inclusion statement: In conducting this research, we adhered to ethical principles and embraced inclusivity throughout the study. The study received approval from the local ethics review committee, and participant safety and well-being were paramount, with provisions in place to address potential risks. R.L.G. is a co-founder, scientific advisor, consultant, and equity holder of MatriSys Biosciences and is a consultant who receives income and equity in Sente. R.K. is a scientific advisory board member, and consultant for BiomeSense, Inc., has equity, and receives income. He is a scientific advisory board member and has equity in GenCirq. He is a consultant and scientific advisory board member for DayTwo, and receives income. He has equity in and acts as a consultant for Cybele. He is a co-founder of Biota, Inc., and has equity. He is a cofounder of Micronoma, and has equity and is a scientific advisory board member. The remaining authors declare no competing interests.
