## [Peer Review File · Nature Communications]

Dermal injury drives a skin to gut axis that disrupts the intestinal microbiome and intestinal immune homeostasis in miceREVIEWER COMMENTS

Reviewer #1 (Remarks to the Author):

In the manuscript by Dokoshi et al, the authors discovered that skin homeostasis imbalance can induce an alteration of intestinal host defenses and gut microbiota.

The authors have previously found that the skin inflammation that follows wounding induces the differentiation of a subset of fibroblasts in the colon into preadipocytes. This is mediated by the recognition of digested HA (hyaluronan), produced during skin damage, by TLR4 on the intestinal fibroblasts. The increase in preadipocytes correlates with an increased susceptibility to DSS-induced colitis (Dokoshi et al, JCI, 2021).

Inflammatory disorders affecting epithelial tissues often co-occur (e.g., patients with psoriasis have a higher risk of IBD). This seems to indicate that alterations of skin homeostasis can have pathologic effects in distal epithelial barrier tissues, such as the intestine. However, the mechanisms underlying this crosstalk are poorly understood. In the current study, the authors have built on their previous study and provided another potential mechanistic explanation as to why skin damage and inflammation exacerbate DSS-induced colitis. Following skin damage by incisional wound or keratinocyte-specific hyaluronidase over-expression, they observed an increased expression of antimicrobial genes (i.e., Muc2, Reg2b and Reg3d) and increased expression of TNF in the intestinal tract. Consistently with this, they found an overall reduction of live bacteria. However, the authors observed an increased presence of some pathogenic bacteria with an increased capacity of penetrating the colonic epithelium. Therefore, they concluded that skin damage has a significant effect on the composition of the gut microbiota. Finally, using antibiotic and fecal transplantation the authors showed that the increased susceptibility of mice to DSS-induced colitis, due to skin damage, depends on the microbiome. Hence, the microbiome alterations caused by skin damage-induced changes in intestinal host defenses, render the intestine more susceptible to damaged-induced inflammation.

These findings are therefore very relevant because they provide mechanistic evidence for the existence of a skin-intestine communication axis and have potential therapeutic implications for patients suffering from psoriasis and IBD.

Overall, the experiments are well performed, with the most appropriate controls. The results are clearly presented and conclusions justified. However, some additional

experiments would be required to strengthen the study.

Comments to address:

1) Figure 2A, 2D and 2F, please provide staining quantifications of sections of the different experimental groups and provide higher resolution images. It is really hard at the moment to determine how specific is the staining. Please also provide an inset for each representative image, to have a higher magnification view of the stained cells.

2) Figure 4D, please provide staining quantification.

3) The authors observed increased mRNA levels in the intestines of wounded or K14/HYAL1 transgenic, DSS-treated mice compared to control DSS treated mice. This increase depends on the presence of intestinal bacteria since it is no longer observed in germ-free mice or mice treated with antibiotics.

Is TNF produced by the intestinal epithelial cells? Does TNF have a direct role in the increased susceptibility to DSS-induced damage observed in the wounded or K14/HYAL1 mice? Or the augmented production of TNF is only the consequence of increased damage? If possible, the authors should perform an experiment where DSS-treated control and K14/HYAL1 mice are injected with NaCl (control) or TNF blocker (e.g. Etanercept). Given that TNF can induce cell death, the authors could do TUNEL staining on intestine sections to address whether the increased damaged caused by DSS in wounded or K14/HYAL1 mice correlates with cell death amounts.

Minor comments:

1) Higher resolution figures should be provided, it is very hard to read the figures in their current resolution.

2) The authors should refer to their previous findings (Dokoshi et al, JCI, 2021) both in the introduction and in the discussion, and comment whether they think there is a relationship between the skin damage-induced fibroblasts-to-adipocytes differentiation and the

intestinal host defenses and microbiome changes. In other words, do they think these two distinct phenomena, both originating from skin damage, equally or differentially contribute to the susceptibility to DSS-induced colitis?

Reviewer #2 (Remarks to the Author):

The manuscript by Dokoshi T et al. "Dermal injury drives a skin-gut axis that disrupts the intestinal microbiome and intestinal immune homeostasis" assesses the role of skin injuries in the disruption of intestinal microbiome. The authors propose novel perspective of skin-gut axis actions by reporting that skin injuries can also cause the changes in the gut microbiome composition and by this increase in susceptibility to intestinal inflammation when exposed to a trigger such as DSS. This report provides interesting explanation for the co-occurrence of skin and intestinal diseases in patients. The methodology applied in the work is sound. Authors used different models and approaches models to support their conclusions. The experiments are carried out well. However, some structures require improvement. The following suggestions can be considered by authors to improve the quality of their manuscript.

The Material and Methods section is missing information that are prerequisite for ensuring reproducibility of the experiments. Can the authors please provide missing information - housing conditions and hygiene status of animals used in the study, the strain and source of used GF mice should be also given, reference 15 is not giving the detailed description of disease activity index. Please provide information whether and which humane endpoint criteria were used for the survival study. Furthermore, no detailed explanations of a method are given in reference 40, and 41 - please cite the reference with detailed description or give the method description. In the mouse model of skin wounding by incision - please give information whether the wound was closed and how, were mice received analgesia and analgesia, how was this procedure performed under GF conditions. For mouse colon ex-vivo model - please give information whether the colon was flushed before treatment, cut in smaller parts, please state incubation conditions. For cell culture please give a brief digestion protocol for production of LMWHA, how was it done. For FMT transfer please state how long the experiment was performed, this is not stated in the manuscript. The table S1 was missing in the supplementary data.

In line 59 is a typo for IBD abbreviation.

In line 66-68, can the authors please add references supporting their claim?

In line 89, the authors state that skin induces expression of Muc2 and Reg3 in the colon, but as the increase of the expression was shown for in the models of skin injuries, would it not be precise to state skin injury?

Can the authors state what they used as control in Fig1? Did they used WT mice on B6 background or WT littermates derived from K14/HYAL1 breeding were used as control?

In line 109 authors state that 18 different clusters were determined and in the S2 Figure the localization of the clusters was shown, but can the authors elaborate on parameters used to group these cell in the clusters e.g. epithelium, crypt (mentioned below in line 112).

The authors argue that the loss of viability of some bacteria is in line with increased Muc2 and Reg3 expression. To my knowledge, only Reg3 are described to have bactericidal activity, and Mu2 is the major gel-forming mucin of the colon mucus. Can the authors elaborate on the how Muc2 can decrease bacterial viability?

In Fig. 4, please check the legend a-c as it seems different to the graphs shown in a-c.

Can the authors elaborate bit further, that even without intestinal trigger and with the increased gene expression of Muc2 and Reg3 there is a higher infiltration of bacteria in the intestinal tissue upon skin injury. It would be expected that the due to increased Muc2 production the mucus layer would be thicker and push the cell even further into the lumen, and the increase in Reg3 would eliminate more even bacteria. Have the authors in that line measured the thickness of the mucus or the mucus permeability?

In Fig. 4f showing in situ hybridization of 16S rRNA gene it is very hard to distinguish between specific and unspecific staining. Have the authors thought to include insets with higher magnifications to better visualize the reported observations? The insets with higher magnification would be valuable in all performed staining showing bigger tissue parts.

The authors showed that bacteria are responsible for increased disease severity following DSS treatment. Was this related to the greater infiltration of bacteria into gut tissue, as DSS is known to cause epithelial barrier damage?

In Fig. 5e and 5k, please give "colon length" as Y-axis title, it will increase understandability of the graph.

Can the authors elaborate on the HA transport into the gut in humans? In the paper, they discuss ingestion as a major way. This is probably true in the context of animal models, but

in humans, probably other transport routes would be more important.

How does the skin wound without DSS treatment influence the mRNA expression level of TNF α in SPF mice?

Reviewer #3 (Remarks to the Author):

This is an interesting and important study with novel findings. The model is interesting but also somewhat artificial although I was convinced by some of the data presented that the dermal microbiome does influence the gut via HA however statistics and sample size undermine the conclusions drawn. Skin injury may exacerbate colitis but the histopath does not support this.

There are some remaining concerns that need to be addressed:

- 1) What is the sample size for figure 1i and K?
- 2) Histopathological scoring needs to be done and presented in Figure 1 - it looks like there is less damage not more in the swiss roll. This data is not convincing and 2 blinded scorers are needed to evaluate and quantify this.
- 3) Wherever there is a sample size of 3 this is unacceptable. This is a novel data that indicates the skin is changing gut inflammation. Therefore the data needs to be reproducible and a sample size of 3 is not.
- 4) Multiple corrections have to be accounted for and in no situation the way the design of the experiment has been presented would there ever be a t test. You have non DSS, DSS and multiple genotypes.
- 5) Figure 4 is very dramatic. How can live bacteria decrease yet so much bacteria present in the IECs without any change in barrier function. These results should be substantiated. There are bacteria in the control as well and that in itself is actually very unusual. I suggest to clarify this, the bacteria should be stained in conjunction with the mucin 2 via carnoyl's staining.
- 6) SD not SEM should be presented

Point-by-point response

Reviewer #1 (Remarks to the Author)

In the manuscript by Dokoshi et al, the authors discovered that skin homeostasis imbalance can induce an alteration of intestinal host defenses and gut microbiota.

The authors have previously found that the skin inflammation that follows wounding induces the differentiation of a subset of fibroblasts in the colon into preadipocytes. This is mediated by the recognition of digested HA (hyaluronan), produced during skin damage, by TLR4 on the intestinal fibroblasts. The increase in preadipocytes correlates with an increased susceptibility to DSS-induced colitis (Dokoshi et al, JCI, 2021).

Inflammatory disorders affecting epithelial tissues often co-occur (e.g., patients with psoriasis have a higher risk of IBD). This seems to indicate that alterations of skin homeostasis can have pathologic effects in distal epithelial barrier tissues, such as the intestine. However, the mechanisms underlying this crosstalk are poorly understood. In the current study, the authors have built on their previous study and provided another potential mechanistic explanation as to why skin damage and inflammation exacerbate DSS-induced colitis. Following skin damage by incisional wound or keratinocyte-specific hyaluronidase over-expression, they observed an increased expression of antimicrobial genes (i.e., Muc2, Reg2b and Reg3d) and increased expression of TNF in the intestinal tract. Consistently with this, they found an overall reduction of live bacteria. However, the authors observed an increased presence of some pathogenic bacteria with an increased capacity of penetrating the colonic epithelium. Therefore, they concluded that skin damage has a significant effect on the composition of the gut microbiota. Finally, using antibiotic and fecal transplantation the authors showed that the increased susceptibility of mice to DSS-induced colitis, due to skin damage, depends on the microbiome. Hence, the microbiome alterations caused by skin damage-induced changes in intestinal host defenses, render the intestine more susceptible to damaged-induced inflammation.

These findings are therefore very relevant because they provide mechanistic evidence for the existence of a skin-intestine communication axis and have potential therapeutic implications for patients suffering from psoriasis and IBD.

Overall, the experiments are well performed, with the most appropriate controls. The results are clearly presented and conclusions justified. However, some additional experiments would be required to strengthen the study.

Response: Thank you for the positive review of our work, we have performed the additional experiments and analysis as described below.

Comments to address:

1) Figure 2A, 2D and 2F, please provide staining quantifications of sections of the different experimental groups and provide higher resolution images. It is really hard at the moment to determine

how specific is the staining. Please also provide an inset for each representative image, to have a higher magnification view of the stained cells.

Response: We have incorporated higher magnification images and staining quantifications for Figure 2A, 2D, and 2F by adding Figure S4a (enhanced magnification of PAS staining), and Figure S4b (quantitative data on mucin staining).

Moreover, to ensure a more detailed view of stained cells, we have included insets (Figure 2e and Figure S4c).

2) Figure 4D, please provide staining quantification.

Response: We added quantification of the gram staining shown in Figure 4d as a new Figure 4e. We also provide new data in Figure S8 a, b that shows additional section orientation and semi quantitative scoring of Gram stain in crypts and muscle layers. The detailed methods are provided in methods based on similar prior semi-quantitative methods for assessing gram staining.

3) The authors observed increased mRNA levels in the intestines of wounded or K14/HYAL1 transgenic, DSS-treated mice compared to control DSS treated mice. This increase depends on the presence of intestinal bacteria since it is no longer observed in germ-free mice or mice treated with antibiotics.

Is TNF produced by the intestinal epithelial cells? Does TNF have a direct role in the increased susceptibility to DSS-induced damage observed in the wounded or K14/HYAL1 mice? Or the augmented production of TNF is only the consequence of increased damage? If possible, the authors should perform an experiment where DSS-treated control and K14/HYAL1 mice are injected with NaCl (control) or TNF blocker (e.g. Etanercept).

Given that TNF can induce cell death, the authors could do TUNEL staining on intestine sections to address whether the increased damaged caused by DSS in wounded or K14/HYAL1 mice correlates with cell death amounts.

Response: We appreciate the reviewer's insightful feedback. In response to the request, we performed new experiments to test if the severe colitis after skin injury or K14/HYAL1 mice correlates with TNF production and if TNF is important to disease outcome. We now show that cell death is enhanced in skin wound or K14/HYAL1 mice by TUNEL staining in Figure S10a,

b. We also tested if the administration of neutralizing antibodies to TNF (XT3.11) could rescue colitis seen after skin wounding. Figure S10c- e shows that TNF is increased in the colon after DSS and skin wounding when compared to DSS without skin wounding (Fig S10c) and that anti-TNF treatment abrogates the increase in TNF and cell inflammation induced by skin wounding (Fig S10c-e). Therefore, we conclude that TNF is important and appears to have a direct role in increased susceptibility to DSS-induced damage observed in the wounded or K14/HYAL1 mice, but the presence of bacteria in the gut is necessary to induce TNF (see edits pg6-7).

Minor comments:

1) Higher resolution figures should be provided, it is very hard to read the figures in their current resolution.

Response: We are providing higher resolution figures.

2) The authors should refer to their previous findings (Dokoshi et al, JCI, 2021) both in the introduction and in the discussion, and comment whether they think there is a relationship between the skin damage-induced fibroblasts-to-adipocytes differentiation and the intestinal host defenses and microbiome changes. In other words, do they think these two distinct phenomena, both originating from skin damage, equally or differentially contribute to the susceptibility to DSS-induced colitis?

Response: We refer to our previous findings in the second paragraph of the introduction and throughout the discussion. Our results suggest that while both events take place, increased severity of DSS colitis due to the skin is dependent on the presence of bacteria. Increased adipogenesis is also dependent on colitis triggered by bacteria and augmented by HA. However, we cannot say if these are distinct phenomena since we do not have evidence that the adipogenesis response of fibroblasts to skin HA has influence on the epithelial Reg3 and Muc2 that likely leads to the change in the function of the bacteria in the gut. We have modified our discussion on page 8 to address this issue.

Reviewer #2 (Remarks to the Author)

The manuscript by Dokoshi T et al. “Dermal injury drives a skin-gut axis that disrupts the intestinal microbiome and intestinal immune homeostasis” assesses the role of skin injuries in the disruption of intestinal microbiome. The authors propose novel perspective of skin-gut axis actions by reporting that skin injuries can also cause the changes in the gut microbiome composition and by this increase in susceptibility to intestinal inflammation when exposed to a trigger such as DSS. This report provides interesting explanation for the co-occurrence of skin and intestinal diseases in patients. The methodology applied in the work is sound. Authors used different models and approaches models to

support their conclusions. The experiments are carried out well. However, some structures require improvement. The following suggestions can be considered by authors to improve the quality of their manuscript.

Response: Thank you for the positive review of our work, we revised our manuscript based on the important points you have raised.

1, The Material and Methods section is missing information that are prerequisite for ensuring reproducibility of the experiments. Can the authors please provide missing information - housing conditions and hygiene status of animals used in the study, the strain and source of used GF mice should be also given, reference 15 is not giving the detailed description of disease activity index. Please provide information whether and which humane endpoint criteria were used for the survival study. Furthermore, no detailed explanations of a method are given in reference 40, and 41 - please cite the reference with detailed description or give the method description. In the mouse model of skin wounding by incision - please give information whether the wound was closed and how, were mice received analgesia and analgesia, how was this procedure performed under GF conditions. For mouse colon ex-vivo model - please give information whether the colon was flushed before treatment, cut in smaller parts, please state incubation conditions. For cell culture please give a brief digestion protocol for production of LMWHA, how was it done. For FMT transfer please state how long the experiment was performed, this is not stated in the manuscript. The table S1 was missing in the supplementary data.

Response: We apologize for any confusion or unintended omissions in providing detailed description of our methods.

The mouse housing information is now described in more detail in Animal and animal care and microbiome analysis sections.

We added the original reference for histological analysis as new reference 41.

For the survival study we have added the statement in methods that mice that lose more than 20 % of their original body weight were euthanized.

We have expanded the detail in description of wounding, anesthesia and mouse care in germ-free facility.

For ex-Vivo experiment, we have revised description in the section *Mouse colon ex vivo model* For LWHHA, please see expanded methods in Cell Culture section.

2, In line 59 is a typo for IBD abbreviation.

Response: Thank you for noticing this, it has been corrected

3, In line 66-68, can the authors please add references supporting their claim?

Response: We appreciate the reviewer's feedback. In order to address this comment, we added the references 14 and 15 here.

4, In line 89, the authors state that skin induces expression of Muc2 and Reg3 in the colon, but as the increase of the expression was shown for in the models of skin injuries, would it not be precise to state skin injury?

Response: We appreciate the reviewer's feedback and agree. In order to address this comment, we changed the title to "Hyaluronidase activity in the skin induces expression of Muc2 and Reg3 in the colon"

5, Can the authors state what they used as control in Fig1? Did they used WT mice on B6 background or WT littermates derived from K14/HYAL1 breeding were used as control?

In line 109 authors state that 18 different clusters were determined and in the S2 Figure the localization of the clusters was shown, but can the authors elaborate on parameters used to group these cell in the clusters e.g. epithelium, crypt (mentioned below in line 112).

The authors argue that the loss of viability of some bacteria is in line with increased Muc2 and Reg3 expression. To my knowledge, only Reg3 are described to have bactericidal activity, and Mu2 is the major gel-forming mucin of the colon mucus. Can the authors elaborate on the how Muc2 can decrease bacterial viability?

Response: We used WT littermates derived from K14/HYAL1 breeding were used as control in Figure 1 and have added this statement to the legend to clarify. To clarify cluster information, we modified Figure 1a and added figure S2 to better illustrate anatomical information.

In response to the reviewer's inquiry about the claim that the loss of viability of certain bacteria aligns with increased Muc2 and Reg3 expression, we acknowledge and appreciate the opportunity to provide clarification. While we recognize that Muc2 is primarily known for its role as a major gel-forming mucin in the colon mucus, our intention was not to imply a direct bactericidal activity attributed to Muc2. Instead, our interpretation aligns with the notion that both Muc2 and Reg3 serve as protective molecules against bacterial invasion in the epithelial barrier.

To address this concern explicitly, we have revised the manuscript to convey our perspective more accurately. We have emphasized that while Muc2 is not recognized for its direct bactericidal properties, its presence contributes to the overall host defense mechanism by maintaining a robust physical barrier. This synergy between the mucin network and bactericidal activity, primarily attributed to Reg3, collectively fortifies the epithelial barrier against bacterial infiltration.

6, In Fig. 4, please check the legend a-c as it seems different to the graphs shown in a-c. Can the authors elaborate bit further, that even without intestinal trigger and with the increased gene expression of Muc2 and Reg3 there is a higher infiltration of bacteria in the intestinal tissue upon skin injury. It would be expected that due to increased Muc2 production the mucus layer would be thicker and push the cell even further into the lumen, and the increase in Reg3 would eliminate more even bacteria. Have the authors in that line measured the thickness of the mucus or the mucus permeability?

Response: Thank you for noting this error and we have fixed Fig4 a-c legends.

We completely agree with your expectation that increased Muc2 and Reg3 could decrease bacteria, and we show decreased live bacteria and 16S in feces in Fig4a-c consistent with this comment. We evaluated the epithelial permeability in Figure 4h and found no change by wounding or K14/HYAL1. The unexpected and exciting observation was that our analysis suggested that these surviving microbes could better penetrate the epithelium as seen by gram stain and 16S (Fig 4d-g) and metagenomics that showed the presence of genes in surviving bacteria that encode processes that are associated with bacterial survival. We have expanded comments of this point in the discussion (lines 272-276).

7, In Fig. 4f showing in situ hybridization of 16S rRNA gene it is very hard to distinguish between specific and unspecific staining. Have the authors thought to include insets with higher magnifications to better visualize the reported observations? The insets with higher magnification would be valuable in all performed staining showing bigger tissue parts. The authors showed that bacteria are responsible for increased disease severity following DSS treatment. Was this related to the greater infiltration of bacteria into gut tissue, as DSS is known to cause epithelial barrier damage?

Response: To better visualize our observations, we have included insets with higher magnifications, particularly focusing on crypt structures (see Figure S8 and in Fig 2e).

Regarding the association between bacterial infiltration and increased disease severity following DSS treatment, our data strongly support this connection. We observed a correlation between bacterial infiltration, severe inflammation, and the depletion of the gut microbiome in both antibiotic-treated and germ-free conditions, resulting in diminished colitis severity after skin injury. This aligns with the known impact of DSS in causing epithelial barrier damage, further emphasizing the crucial role of bacterial infiltration in the exacerbated colitis observed in our study. However, although we show increased permeability after DSS in Fig 4h, we did not see a

change in permeability by skin wounding or K14/HYAL1 but did see increased bacterial penetration in the colon after skin wounding or K14/Hyal1 without DSS challenge. Therefore, we conclude that the increase in bacterial penetration was a consequence of the shift in the microbiome.

8, In Fig. 5e and 5k, please give “colon length” as Y-axis title, it will increase understandability of the graph.

Response: We have changed Figure 5e Y- axis to read “colon length (cm)”

9, Can the authors elaborate on the HA transport into the gut in humans? In the paper, they discuss ingestion as a major way. This is probably true in the context of animal models, but in humans, probably other transport routes would be more important.

How does the skin wound without DSS treatment influence the mRNA expression level of TNF α in SPF mice?

Response: We appreciate the reviewer's feedback. Our animal model induces colitis through ingestion of DSS, but HA transport is not mediated through ingestion. In our previous study (DOI: 10.1172/JCI147614) we showed elevated HA levels in both serum and the colon after skin injury, indicating that the bloodstream is probably a crucial transport route in the context of this tissue interaction. The DSS ingestion model as well as our previously published results in spontaneous colitis in the IL10^{-/-} mouse model enabled detection of the effects of increased HA in the circulation.

In response to the query about the influence of TNF α in SPF mice, we have now included the TNF levels from the skin wound-only group in Figure 5b. Our observations indicate that TNF levels are not significantly affected in the skin wound-only group and remain consistent with other non-DSS-treated samples. This underscores that the impact on TNF α expression is specific to the DSS-treated conditions, and the skin wound alone does not induce a substantial change in TNF α mRNA expression in the absence of both bacteria and DSS challenge. Furthermore, we now also present results from new experiments with TNF neutralization that show TNF is critical to the increase in inflammation that is driven by skin wounds (Fig S10).

Reviewer #3 (Remarks to the Author):

This is an interesting and important study with novel findings. The model is interesting but also

somewhat artificial although i was convinced by some of the data presented that the dermal microbiome does influence the gut via HA however statistics and sample size undermine the conclusions drawn. Skin injury may exacerbate colitis but the histopath does not support this.

There are some remaining concerns that need to be addressed:

1) What is the sample size for figure 1i and K?

Response: For spatial transcriptomics the sample size is determined by the number of sequencing spots present under the tissue. Specifically, the counts are as follows: WT (4395 sequencing spots), DSS (3813 sequencing spots), K14/HYAL1 (3533 sequencing spots), and K14DSS (3397 sequencing spots). This information has been added to the Figure legend.

2) Histopathological scoring needs to be done and presented in Figure 1 - it looks like there is less damage not more in the swiss roll. This data is not convincing and 2 blinded scorers are needed to evaluate and quantify this.

Response: Figure 1 presents spatial sequencing results and does not intend to provide information on histological damage. Please note that the capacity of skin inflammation to increase colitis from DSS or in IL10^{-/-} mice was shown previously in Dokoshi et al JCI 2021. Figure 5 provides H&E images of the colon to illustrate tissue damage and the capacity of vancomycin to rescue, or FMT to transfer increased injury. As the reviewer suggested we have now performed histopathological scoring by two blinded scorers and present scoring data on disease severity in Figure 5g, Figure 5m and on bacterial penetration in Figure S8 .

3) Wherever there is a sample size of 3 this is unacceptable. This is a novel data that indicates the skin is changing gut inflammation. Therefore the data needs to be reproducible and a sample size of 3 is not.

Response: We appreciate the reviewer's feedback and agree that increased sample size will provide greater confidence regarding the reproducibility of these important observations. To address this concern, we have now assured that all functional experiments have been done with a minimum sample size of 5. Functional experiments were also replicated at least 3 times. In the case of microbiome analysis, the sample size was 32.

4) Multiple corrections have to be accounted for and in no situation the way the design of the experiment has been presented would there ever be a t test. You have non DSS, DSS and multiple genotypes.

Response: Thank you for noting this and we agree that other more rigorous statistical tests are more appropriate in several of the experiments. We have therefore corrected the statistical analysis and when appropriate changed to one-way ANOVA with the appropriate multiple comparison corrections. This is now detailed in each figure legend. This change in the testing method did not result in any change in our conclusions.

5) Figure 4 is very dramatic. How can live bacteria decrease yet so much bacteria present in the IECs without any change in barrier function. These results should be substantiated. There are bacteria in the control as well and that in itself is actually very unusual. I suggest to clarify this, the bacteria should be stained in conjunction with the mucin 2 via carnoyl's staining.

Response: We appreciate the reviewer's feedback. Our samples are treated for carnoyl's method and Mucin staining is shown in Fig 2 a. As you point out, almost no bacteria are seen by gram staining in the control sample. Furthermore, to substantiate these observations, independent experiments were done by using in situ hybridization with probes specific for the 16S DNA found in bacteria. These results confirmed the results by gram staining and showed a large increase of bacterial 16s deep in the crypt. Both the gram staining and 16S hybridization results are now also quantified by ImageJ in Fig 4e and g.

6)SD not SEM should be presented

Response: The data in all the figures are now shown as SD.

REVIEWERS' COMMENTS

Reviewer #1 (Remarks to the Author):

The authors addressed all the points of criticism raised by this reviewer and certainly increased the quality and robustness of the manuscript.

Reviewer #2 (Remarks to the Author):

The Authors have addressed all of my concerns with the original manuscript. The revised manuscript can be accepted for publication.

Reviewer #3 (Remarks to the Author):

I am satisfied with the response to the reviews as queries have been adequately addressed.

Point by Point response

REVIEWERS' COMMENTS

Reviewer #1 (Remarks to the Author):

The authors addressed all the points of criticism raised by this reviewer and certainly increased the quality and robustness of the manuscript.

Reviewer #2 (Remarks to the Author):

The Authors have addressed all of my concerns with the original manuscript. The revised manuscript can be accepted for publication.

Reviewer #3 (Remarks to the Author):

I am satisfied with the response to the reviews as queries have been adequately addressed.

RESPONSE

We thank all the reviewers for their helpful comments and appreciate their part in improving this report.